# Unlocking the Potential of Fish to Improve Food and Nutrition Security in Sub-Saharan Africa

Rodney T. Muringai [1,*], Paramu Mafongoya [2], Romano T. Lottering [3], Raymond Mugandani [4] and Denver Naidoo [1]

1   African Centre for Food Security (ACFS), School of Agriculture Earth and Environmental Sciences, University of KwaZulu Natal, Pietermaritzburg 3201, South Africa; naidook12@ukzn.ac.za
2   Centre for Agriculture and Environmental Development, School of Agriculture Earth and Environmental Sciences, University of KwaZulu Natal, Pietermaritzburg 3201, South Africa; mafongoya@ukzn.ac.za
3   Geography Department, School of Agricultural Earth and Environmental Sciences, University of KwaZulu Natal, Pietermaritzburg 3201, South Africa; lottering@gmail.com
4   Department of Land and Water Resources, Faculty of Agriculture, Environment and Natural Resources, Midlands State University, Gweru, Zimbabwe; mugandanir@gmail.com
*   Correspondence: rodneymuringai@gmail.com

**Abstract:** Approximately one-third of the global population suffering from chronic hunger are in sub-Saharan Africa (SSA). In addition to high prevalence of chronic hunger, millions of people suffer from micronutrient deficiencies. Meanwhile, there is growing consensus across scientific disciplines concurring that fish plays a crucial role in improving food and nutrition security. Therefore, the present review aims to demonstrate the role of fish and the whole fisheries sector towards securing food and nutrition security in SSA by summarizing the existing literature. Fish is a treasure store of animal protein and essential micronutrients such as zinc, iodine, calcium, and vitamins, which are essential in human nutrition and have proven to help reduce the risks of both malnutrition and non-communicable diseases. Policymakers, development agencies, and society should recognize the role that the fisheries sector can play in combatting hunger and undernutrition, especially for the poor and marginalized people in SSA.

**Keywords:** undernutrition; small indigenous species; fish nutrition; sustainable development goals (SDGs); sub-Saharan Africa (SSA)





## 1. Introduction

The issue of food and nutrition security is one of the top global political agendas, standing as goal number two of the Sustainable Development Goals 2030. The goal aims to end hunger and achieve food security and improve nutrition by 2030. The Committee on World Food Security [1] defines food and nutrition security as a situation that exists when all people at all times have physical, social, and economic access to food of sufficient quanitity and quality in terms of variety, diversity, nutrient content, and safety to meet their dietary needs and food preferences for an active healthy life coupled with a sanitary environment and adequate health, education, and care. In the last decade of the realization of the SDGs 2030, the African continent is still inundated with the challenges of poverty and food and nutrition insecurity. Africa is off track in terms of achieving the target of SDG number two which aims to alleviate global hunger by 50% by the year 2030 [2]. At present, more than 100 million people in Africa face catastrophic food insecurity levels [3]. The human population on the African continent is projected to increase to a billion by 2050. The fastest population growth is expected to be in the sub-Saharan Africa region where the prevalence of food and nutrition security has already been documented. Food and nutrition insecurity is a complex phenomenon that requires sound policies that promote food production and trade and exploration of other food production systems that can contribute to the existing

food supplies. In this case, the fisheries sector is one of the food production systems that has been recognized for its significant contribution to promoting socioeconomic growth, enhancing food and nutrition security, and improving the livelihoods of marginalized communities [4,5].

With more emphasis on the nutritional aspect and supply of food commodities, fish is recognized as a major nutrient-dense animal source of food for many nutritionally vulnerable people, surpassing that of most terrestrial animal foods [4]. Fish is the main or only source of animal protein for more than 30% of Africa's human population of more than 200 million people [6,7]. Fish muscle constitutes about 16–21% protein, particularly the essential amino acids methionine and lysine [8]. In addition to protein, fish is a good source of almost all bioavailable minerals, including calcium, iodine, iron, zinc, phosphorus, and selenium fluorine [8]. Besides the nutritional value of fish, the fisheries sector also contributes significantly to livelihoods and economic growth [4,9]. The sector supports the livelihoods of approximately 12.3 million people and contributes about 1.3% share to Africa's Gross Domestic Product (GDP) [5]. The African fisheries sector encompasses marine capture fisheries, inland freshwater capture fisheries, and aquaculture. Nevertheless, within this sector, aquaculture is the fastest-growing sub-sector to the extent that one fish in every two consumed is from aquaculture [10]. The aquaculture sector is rapidly growing because of the increasing fish demand, declining capture fisheries production, and the advent of fish farming technologies. Comparable with other continents, fish demand in Africa has increased due to population and income growth, diet transformation towards increasing demand for protein-rich foods such as meat and fish, and increasing appreciation of health benefits of fish consumption [6]. Growth in fish consumption was high (about 25–50%) in most SSA countries between 2007 and 2015 [6]. Nevertheless, the capacity of the fisheries sector to support livelihoods and provide for food security is under substantial threat due to habitat degradation, illegal fishing, overfishing, poor governance, and climate change [11–13].

Despite the significant contribution of fisheries to livelihoods and food and nutrition security, the High-level Panel of Experts (HLPE) states that the sector has been arbitrarily separated from other parts of food production systems in food security studies, debates, and policy designing [14]. The importance of fisheries for food and nutrition security has often been unrecognized by both the food security community, which is still geared mainly towards food access and availability with a focus on staple foods, and fisheries managers, who mostly focus on the management of fish resource rather than its contribution to people's well-being [14]. This is mainly so because fish are seen as a natural resource, not food, which puts more emphasis on conservation and species management objectives [15]. Therefore, to encourage the creation of policies that support the contribution of fish to food and food nutrition security and shift from viewing fish as a natural resource, it is crucial to demonstrate the potential of fish and the fisheries sector to enhance livelihoods and food and nutrition security.

On that backdrop, this review article aims to demonstrate the role of fish and the whole fisheries sector towards securing food and nutrition security in SSA by presenting the breadth of evidence available to inform and steer policy, investments, and research and creating awareness of the benefits of fish to human nutrition and meeting sustainable development. Special attention is given to the SSA region because (i) population growth is projected to grow rapidly in this region than any other region resulting in increased food demand, (ii) the region is already characterized by high rates of undernourished people, (iii) the region is endowed with vast fishery resources from marine to inland freshwater that can support the fisheries sector and, (iv) there is limited literature on the role of the fisheries sector in addressing food and nutrition security in this region in particular. Clear communication and analysis on the role that fish can play in food and nutrition security maximize the potential of fish to be fused in national or regional policies. However, this study's findings and recommendations are specific to SSA, and they cannot be widespread to other regions.

## 2. Sub-Saharan Africa's Food and Nutrition Status, Causes, and Consequences

In 2016, about 10.7% (815 million people) of the 7.6 billion people were suffering from chronic undernourishment globally [16]. The prevalence of food insecurity is much higher in Africa than in other continents [17]. According to Abegaz [16], Africa has the highest prevalence of undernourishment (PoU) accounting for 25% of the 815 million undernourished people in the world. Moreover, Thome et al. [18] state that more than 50% of Africans are exposed to moderate or severe food insecurity. Despite several global efforts implemented to eradicate hunger, the number of undernourished people in Africa increased from 216.9 million people in 2015 to 250.3 million people in 2019 and projected to further increase to 433.2 million people by 2030 as shown in Table 1 [19].

**Table 1.** Number of undernourished people in Africa from 2005–2030.

| Number of Undernourished People (Millions) | | | | | | | | |
|---|---|---|---|---|---|---|---|---|
| | 2005 | 2010 | 2015 | 2016 | 2017 | 2018 | 2019 * | 2030 * |
| Africa | 192.6 | 196.1 | 216.9 | 224.9 | 231.7 | 236.8 | 250.3 | 433.2 |
| Sub-Saharan Africa | 174.3 | 178.3 | 203.0 | 210.5 | 216.3 | 221.8 | 234.7 | 411.8 |
| East Africa | 95.0 | 98.1 | 104.9 | 108.4 | 110.4 | 112.9 | 117.9 | 191.6 |
| Central Africa | 39.7 | 40.0 | 43.5 | 45.8 | 47.2 | 49.1 | 51.9 | 90.5 |
| Southern Africa | 2.9 | 3.2 | 4.4 | 5.1 | 4.5 | 5.2 | 5.6 | 11.0 |
| Western Africa | 36.9 | 37.0 | 50.3 | 51.2 | 54.2 | 54.7 | 59.4 | 118.8 |

Note: * Projected values (adopted and modified from Otekunrin et al. [19]).

However, the prevalence of undernourishment varies significantly between countries within the region. Within Africa, the majority of the undernourished people are in SSA [20]; about 234.7 million of the 250.3 million undernourished people are in SSA [19]. The highest number of undernourished people was witnessed in East and West Africa which both reported approximately 118 million undernourished people in 2019 [21]. Southern Africa records the lowest number of undernourished people in the SSA region [19].

The FAO [2] and Gezimua [17] identified extreme poverty, population growth, overdependence on rainfed agriculture, poor infrastructure, conflict, climate extremes, and economic slowdowns and downturns as the key drivers of the rise in food insecurity in SSA. These factors affect all dimensions of food security (availability, access, utilization, and stability) through affecting the purchasing power of poor households, food production and supply, and food distribution. For instance, a study by Anderson et al. [22] found that the rise of food insecurity across SSA that started in 2014 was attributed to conflict, specifically in Nigeria and South Sudan. In addition, overdependence on rainfed agriculture in the face of declining rainfall and increased occurrence of droughts in SSA has jeopardized the food and nutrition security in the region.

The increasing challenge of food insecurity in SSA has and will continue to cause devastating impacts on the population and economies at large. For instance, almost half of the 10.9 million child deaths recorded in Africa are attributed to poor nutrition [23]. Nutrient deficiencies during infancy and childhood lead to impaired growth, sub-optimal intellectual development, increased risk of morbidity and mortality, and reduced productivity later in life [24,25], jeopardizing the ability of the future generation to fight poverty and hunger. Bain et al. [25] indicate that undernutrition can result in countries losing about 2–3% of their GDP which exacerbates poverty. Undernourished pregnant women usually give birth to children with low birth weights, consequently growing into mentally and physically stunted children [25]. Furthermore, Gebremedhin and Bekele [26] state that non-communicable diseases (NCDs) are primarily attributed to malnutrition and undernutrition (dietary factors), hence, more than 50% of Africa's population are at high risk of suffering with NCDs. To prevent NCDs, the World Health Organization (WHO) and FAO recommended optimal macronutrient intake ranges based on contribution to daily energy intake of 55–75% carbohydrates, 15–30% fat, and 7–20% protein [27]. However, a study by Gebremedhin and Bekele [26] on the African food supply against the nutrient intake

goals set for preventing diet-related NCDs found that in 2017 energy derived from protein remained below 10% in SSA region despite the notable increase in supply between 1990 and 2017 (Table 2). Consumption of fish can significantly contribute to the WHO and FAO recommended optimal protein and micronutrients intake as fish is a rich source of animal protein and other essential micronutrients. Thus, fish consumption can significantly help to reduce undernutrition and its associated diseases for a better healthy life.

**Table 2.** Trends in protein supply (g/capita/day) in Africa between 1990–2017.

| Protein Supply (g/Capita/Day) | Year | | | | | | | % Change |
|---|---|---|---|---|---|---|---|---|
| | **1990** | **1995** | **2000** | **2005** | **2010** | **2015** | **2017** | |
| World | 70.5 | 72.6 | 75.0 | 76.0 | 80.1 | 81.9 | 82.7 | 31.3 |
| Africa | 66.6 | 67.6 | 71.0 | 74.8 | 78.1 | 80.5 | 79.3 | 19.0 |
| Sub-Sahara Africa | | | | | | | | |
| Central | 57.6 | 55.3 | 66.2 | 66.5 | 72.9 | 83.5 | 80.3 | 39.4 |
| Eastern | 55.o | 53.7 | 55.0 | 58.5 | 61.1 | 66.5 | 66.2 | 20.4 |
| Southern | 82.9 | 81.2 | 83.6 | 86.8 | 86.9 | 88.0 | 87.9 | 6.0 |
| Western | 65.1 | 68.6 | 73.1 | 77.8 | 81.1 | 79.2 | 77.9 | 19.7 |

Source: Adopted and modified from Gebremedhin and Bekele [26]; Roser and Ritchie [28]; and FAO [29].

## 3. Fish Production and Consumption Trends in Sub-Saharan Africa

In SSA, fish production has been growing at a faster rate than any other agricultural product due to population growth and growing appreciation of healthy and nutritious fish-based food [7,30]. According to the FAO [2], Africa's total fish production has increased in the past few decades reaching about 11.8 million tons of fish produced in 2017. However, the contribution of African countries to total fish production varies greatly among countries due to their geographic locations and availability of water resources. For instance, some countries are coastal countries with access to marine fish resources and some countries are landlocked endowed with great lakes and river basins which are highly productive. South Africa, Nigeria, and Uganda are the top capture fishery producers while Nigeria, Uganda, and Ghana are the top aquaculture producers in SSA [31,32]. However, in recent years, capture fisheries have been stagnant due to several reasons such as overexploitation, use of destructive fishing methods, and climate change [12,13,33]. Therefore, the observed increase in the region's fish production might be attributed to the development of the aquaculture sector. The aquaculture sector is acknowledged as the fastest food-producing sector surpassing farm animal meat production and landings from capture fisheries [34]. In Africa alone, the aquaculture sector recorded a twenty-fold production increase from 110,200 to 2,196,000 tons from 1995 to 2018 with a compound rate of about 16% per year, and the contribution by weight to total fish production increased from 6.2% to 18.5% from 2000 to 2017 [31].

According to the United Nations (UN), global per capita fish consumption has more than doubled from approximately 9 kg per capita per annum to about 20.5 kg per capita per annum [35]. Even though fish accounts for more than 30% of total animal protein in Africa, per capita fish consumption within the continent is nearly half of the global average [5,7,35]. In coastal countries in East and West Africa, including Ghana, Sao Tome and Principe, Sierra Leone, Tanzania, and The Gambia, fish accounts for more than 50% of all animal protein consumed and accounts for 30% to 40% in inland countries such as Malawi, Uganda, and Zambia [35]. Per capita fish consumption projections indicate that fish consumption is expected to increase on all continents except in Africa as population growth outpaces fish supply [5,7,36–38]. African fish consumption per capita is expected to drop by 3% from 9.9 kg in 2015–2017 to 9.6 kg in 2027, with a more substantial decrease in SSA [36]. Table 3 indicates low fish consumption per capita levels in most SSA regions indicating underutilization of fish resources especially in regions with high prevalence of undernutrition and where the food and nutrition security situation is worsening [36]. Increasing fish consumption can contribute to the reduction of undernutrition.

Table 3. Per capita fish consumption and fish and animal protein intake by region, 2013.

| Region | Population (Million) | Fish/Animal Protein Intake (%) | Fish Consumption (kg/Person/Year) |
|---|---|---|---|
| World | 6997.3 | 16.2 | 19.0 |
| Africa | 995.4 | 19.3 | 10.8 |
| Central Africa | 67.5 | 26.2 | 14.1 |
| Eastern Africa | 333.0 | 14.7 | 4.8 |
| Northern Africa | 203.2 | 15.0 | 13.5 |
| Southern Africa | 60.4 | 5.2 | 6.1 |
| Western Africa | 331.3 | 34.1 | 15.3 |

Source: Adopted and modified from (Chan et al. [5]).

From a food and nutrition security perspective, the projected decline in African per capita fish consumption, which is a source of valuable micronutrients and protein, will affect millions of malnourished people, particularly the most vulnerable groups (women, children, and the poor). Declining fish intake can impact food and nutrition security and their ability to meet malnutrition targets of the United Nation's SDG 2 which aims to eradicate all forms of hunger and malnutrition by 2030.

## 4. The Contribution of Fish and Fisheries to Food and Nutrition Security

### 4.1. Fish Nutrients Composition

Fish nutrient content depends on species and harvest waters, and thus, nutrient content varies among species. There is limited data on species-specific nutritional composition particularly for species consumed in SSA. However, fish is a rich source of protein and essential nutrients, and there is a growing recognition of its nutritional and health-promoting qualities [14]. Generally, fish muscles constitute about 50–60% of the fish's weight, and the muscle constitutes about 16–21% protein [8,39]. Fish has three main groups of protein which are myofibrillar, sarcoplasmic, and stroma proteins which constitute 70–80%, 20–30%, and 3% of the total muscle protein respectively [39]. Human adults are recommended to take 45–65 g of protein a day and an intake of 100 g of fish can contribute about 15–25% of the total daily protein requirement [38]. In addition, the bioavailability of fish protein is about 5–15% higher than that from plants [9,40].

Fish contains lipids (fatty acids) which have long-chain unsaturated fatty acids (LCPUFA) in the form of arachidonic acid (ARA), eicosapentaenoic acid (EPA), and docosahexaenoic acid (DHA) [9,14,41]. Fish species such as sardines, mackerel, anchovies, and some salmon species are rich in EPA and DHA making them important sources of these fatty acids [38,42]. Nevertheless, the quantity of the total lipids may differ between different fish species and between various tissue organs within the fish depending on their aquatic environment, species feeding habits, spawning period, and physiochemical process [42]. Furthermore, fish is an important source of essential micronutrients such as vitamins A, B, and D [14,40], and bioavailable micronutrients such as, but not limited to, calcium, iodine, iron, zinc, phosphorus, and selenium fluorine [8,43]. The FAO/INFOODS Global Food Composition Database for Fish and Shellfish (uFiSh) provides nutrient data for various fish and shellfish species, covering different catch regions and/or origins of aquaculture production available at http://www.fao.org/3/i6655e/i6655e.pdf (accessed on 7 September 2021) [44]. However, the database only shows the nutrition composition of only 25.7% of the total of 2033 species listed.

It is widely known that small-sized species are an important source of these micronutrients as they are consumed as a whole, with heads and bones [45,46]. Sub-Saharan African waters are endowed with these micronutrient-rich small indigenous species such as the dagaa (*Rastrineobola argentea*) found in Lake Victoria, Usipa (*Engraulicypris sardella*) of Lake Malawi, Kapenta (*Limnothrissa miodon*) found in Lakes Cahora Bassa, Kariba, Kivu, and Tanganyika, and Guimean sprat (*Pellonula leonensis*) in Lakes Kainji and Volta [47]. Thus, the consumption of fish, particularly small indigenous fish species, has a great potential to address the problem of micronutrient deficiencies in people living in the SSA region.

### 4.2. Benefits to Human Nutrition and Health

Fish, particularly small fish, may be the most accessible, affordable, or preferred source of animal protein for many poor or rural populations [9]. Researchers concur that fish consumption has several benefits to human nutrition and health which include but are not limited to, reducing blood pressure, lowering cholesterol levels [48], preventing cancer, and decreasing inflammatory diseases such as arthritis [49], as well as improving maternal and childhood health outcomes, supporting cognitive development, alleviating stunting in children, strengthening the immune system, and reducing cardiovascular disease [14,45,50]. The FAO and World Health Organization [51] argue that the combination of EPA and DHA in fish, oily fish, in particular, lowers the risk of coronary heart disease mortality by up to 36%. Experts stressed that fish consumption reduces mortality due to coronary heart disease in the adult population by 20% [52] and improves the neurodevelopment of fetuses and infants, and is therefore important for women of childbearing age, pregnant women, and nursing mothers [14]. Poly-unsaturated fatty acids in the form of DHA positively influence the optimal brain and neural system development in neonates and infants which is particularly important during pregnancy and the first two years of life [46]. Fish also improves the absorption of micronutrients such as iron and zinc from plant-source foods when consumed together [48].

The contribution of fish to human health and nutrition has been observed in many countries in SSA. For instance, a study by Fiorella et al. [53] on the influence of fish consumption on breast milk acid concentrations in lactating women around Lake Victoria in Kenya found high levels of beneficial breast milk LCPUFA and essential ALA and DHA, exceeding the global and regional averages, when fish was consumed approximately twice within 72 h. Intake of LCPUFA is important for maternal and child nutrition [53]. Moreover, a study by Marinda et al. [45] on the contribution of fish to nutritional status in Zambia found that children who consumed fish had better nutritional outcomes and were less likely to be stunted. Small fish which are abundant in the region have high levels of micronutrients such as vitamin A and B12, iron, and zinc which are critical for child growth and development, brain function, and nervous system maintenance [45]. In Ghana, a study by Akuffo and Quagrainie [54] concluded that fish farming households attained higher nutritional quality than non-fish farming households due to direct consumption of their fish catch. This evidence strongly suggests that fish may have a crucial role in preventing some of the most impactful diseases of modern society and addresses local diet-related diseases, particularly in SSA where there are severe nutrient deficiencies.

### 4.3. Risks to Human Health

Despite being a treasure store for essential nutrients that are beneficial to human health, consumption of contaminated fish poses threats to human health. Fish can contain several hazardous organic and inorganic compounds that pose risk to human health [14]. According to the Scientific and Technical Advisory Panel of the Global Environment Facility (STAP), the compounds found in contaminated fish that present the most significant health hazards are heavy metals such as cadmium, organic tin, and methylmercury [55]. For instance, methylmercury which is neurotoxic is known to have the strongest toxicity to humans which affects the central nervous system in children, the peripheral nervous system in adults causing loss of skin sensation, visual constriction, ataxia and loss of speech and hearing [14,56–58]. In addition, exposure to methylmercury during fetal development may cause severe mental retardation, birth defects, or even fetal death [58]. Studies, for example, Adegbola et al. [59] assessed health risks associated with consumption of fish from polluted Ogun and Eleyele rivers in Nigeria and the study findings show that consumers of fish from the study sites might experience significant non-carcinogenic health risk. In Zimbabwe, a study by Manjengwa et al. [60] found that about 43% of fish consumers reported cases of bloody diarrhea and loose stool after consuming fish from Lake Chivero.

Cyanobacteria, which are found in most inland waters, produce highly toxic secondary metabolites known as cyanotoxins which are hazardous to humans [61]. These toxins which

include nodularins, microcystins, and cylindrospermopsins accumulate in fish tissues and human consumption of contaminated fish can cause gastrointestinal disturbances, skin toxicity, liver, and kidney damage in humans [61,62]. Furthermore, the growth of the aquaculture subsector has been squired by a rapid increasing usage of antibiotics/antimicrobial agents including those used in human therapeutics to defeat the challenges associated to unhealthy and sanitary conditions in aquaculture [14]. There is abundant evidence suggesting that unrestricted use of antibiotics is detrimental to fish and human health [14,63]. However, in spite of the health risks associated with consumption of contaminated fish, experts tend to agree that the positive effects of high fish consumption surpass the potential negative effects associated with contamination risks [14].

### 4.4. Socio-Economic Benefits of Fish and Fisheries

Besides being a good source of immense nutrients that contribute significantly to human nutrition and health, fisheries already play an important social and economic role in Africa. The sector represents a key socioeconomic net through contributions to job creation, generating income, and foreign exchange earnings for several countries [39]. The SSA fisheries sector employs about 12.3 million people of which half of the people are fishers, the other 42.4%, and 7.5% are processors and fish farm workers, respectively (Table 4) [64]. According to de Graaf and Garibaldi [64] inland fisheries employ the majority (40.4%) of fishers and fish processors followed by marine artisanal, then marine industrial and aquaculture, which employs about 32.9%, 19.2%, and 7.5%, respectively (Table 5). While most of the jobs in the fisheries sector are dominated by men, women constitute more than 25% of the people working in the fisheries sector, and the majority (69.2%) of those women work as processors in inland fisheries [64,65]. In SSA, women are involved in post-harvest activities such as fish processing, fish trading, supplying fishing gear, and providing credit [66]. For instance, in Msaka (Lake Malawi) and Kachulu (Lake Chilwa), women dominated as local brokers, 67% in both Msaka and Kachulu beaches, processors, 51% in Msaka beach and 81% in Kachulu beach, and as fish exporters, 100% and 83% in Msaka and Kachulu beaches, respectively [67]. In addition, 80–90% of fish traders in Congo are women [68]. This evidence demonstrates that the fisheries sector plays a significant role in improving livelihoods and empowering women, the poorest and most vulnerable group in developing countries [68]. Women empowerment and improved livelihood strategies enhance fishing households' incomes. A study by Kapembwa et al. [69] confirmed that fishing and its related activities have a bearing on the levels of fishing households' incomes of fishers in Lake Itezhi-Tezhi in Zambia in general. Incomes from fish sales increase the purchasing power resulting in a greater proportion of income being spent on food, enhancing food and nutrition security [9]. Thus, incomes from fisheries and their related activities enhance both households' economic access to food and food and nutrition security. Fisheries also provide a safety net for the poor when other economic opportunities are limited, for instance, Kupaza et al. [70] indicated that more than 80% of fishers in Zimbabwe undertook fishing as a part-time or full-time activity due to high unemployment in the country.

**Table 4.** Distribution of employment in the fisheries and aquaculture sector.

| Type of Work | Number of Employees (Thousands) | Share within the Sector (%) |
|---|---|---|
| Fishers | 6147 | 50.1 |
| Processors | 5202 | 42.4 |
| Fish farm workers | 920 | 7.5 |

Source: Adopted and modified from de Graaf and Garibaldi [64].

**Table 5.** Employment by subsector.

| Subsector | Number of Employees (Thousands) | Share Subsector (%) |
|---|---|---|
| Inland fisheries | 4958 | 40.4 |
| Marine artisanal fisheries | 4041 | 32.9 |
| Marine industrial fisheries | 2350 | 19.2 |
| Aquaculture | 920 | 7.5 |

Source: Adopted and modified from de Graaf and Garibaldi [64].

According to the United Nations Conference on Trade and Development (UNCTAD) [71], fish is one of the most highly traded commodities globally placing aquatic environments (oceans, seas, lakes, rivers, and wetlands, etc.) at the center of economic growth. In Africa, the fisheries sector contributes about 1.6% or USD 24 billion to the continent's GDP [64]. However, the contribution to national GDPs is highly variable across the region and within the region. For instance, in West Africa, fisheries contribute about 2% of the region's total GDP but in Senegal alone (a west-African country) fisheries contribute about 13.5% to the GDP [72], and the sector contributes about 2.7–6.6% in east African countries such as Madagascar, Mozambique, Tanzania, and Zanzibar [64]. However, research indicates that the contribution of fisheries to GDP especially is undervalued due to the nature of the industry; most of the fish produce is consumed or traded locally and does not enter the formal economy and many fisheries operate in remote areas [73,74]. For example, Belhabib et al. [75] found that the contribution of small-scale fisheries to Guinea's GDP was six times higher than the reported estimates. However, economic growth alone does not solve the problem of undernutrition and malnutrition [76]; however, Gillespie et al. [77] argues that a 10% increase in economic growth reduces the challenges of undernutrition and malnutrition by only 6%. Thus, the contribution of fisheries to SSA's economic growth might help to alleviate the challenges of hunger and undernutrition in the region.

## 5. Opportunities and Challenges of Fish for Food and Nutrition Security in Sub-Sharan Africa

As a rich source of animal protein and valuable micronutrients, fish has great potential to address the high levels of undernutrition and malnutrition in SSA. The SSA region is characterized by high levels of undernutrition and malnutrition causing stunted growth in children [45]. Nutrients found in fish can contribute towards the much-needed nutrients from animal source foods especially in children [45]. However, despite being a nutrient treasure store, fish consumption in SSA is still low compared with other regions globally [6,7]. Fish is more affordable than other terrestrial animal sources of protein such as red meat, making it more accessible to the poor [15]. Therefore, increasing fish consumption and securing its supply might lessen undernutrition and malnutrition in the region, giving fishers and fish farmers opportunities to increase production. Furthermore, the observed and projected rapid population growth in SSA will fuel the increasing demand for fish food in the region. The projected increase in fish food demand [5] represents important opportunities for fish sector entrepreneurs to take part in the region's economic development.

Despite the immense potential of fisheries to improve the region's food and nutrition security and sustainable development, the SSA fisheries sector is facing numerous challenges that hinder the ability of the sector to contribute to alleviating hunger and malnutrition. Research indicates that fish catches from wild sources have been declining due to several anthropogenic factors and climate change. Key anthropogenic constraints affecting capture fisheries production in SSA include, but are not limited to, overfishing, habitat destruction, poor governance, and unreported and unregulated fishing [7,78]. For instance, Ababouch and Fipi [72] postulate that about 50% of fish stocks in West Africa are overexploited, with unregulated fishing activities as one of the primary drivers of overexploitation. Overexploitation of fishery resources threatens not just the ecosystems but the socioeconomic condition, particularly the food security condition of the fishing communities. Overexploitation reduces the quantity and quality of available catch, often contributing to poverty and food insecurity [79]. Therefore, to ensure the sustainability

of fishery resources and food and nutrition security for fishing communities, fishing activities should be regulated. Unregulated fishing practices will have detrimental effects on the fishery resources and food security. For example, overfishing, poor governance, and unregulated fishing led to the decline in abundance of the commercially important Chambo (*Oreochromis* spp) in Lake Malombe and Lake Malawi [12,80,81]. Without undervaluing the severe impacts caused by anthropogenic factors on fisheries, climate change is widely acknowledged to be the greatest threat to fish production because it interacts with and amplifies the existing non-climatic stressors [31,80,82]. Several researchers such as Ndebele-Murisa et al. [83]; Potts et al. [84]; Belhabib et al. [85]; Cohen et al. [13]; and Utete et al. [86] agree that climate change is significantly contributing to declines in fish abundance in several African fisheries sources. Although African fish production is being affected by climate change, the magnitude of the effects of climate change on fish stocks are not homogenous across the region. For instance, fish production from African inland fisheries are more prone to climate change as they are more dependent on the external climatic drivers compared with marine fisheries [37]. Changing climatic variables affects fish productivity through changing fish chemicals, geographical distribution, and biological processes such as spawning, metabolism, and reproduction [33,84].

Declining fish productivity has severe effects on food and nutrition security, employment opportunities, and standards of living of fishing households who have limited alternative livelihoods and millions of people who are mainly dependent on fish as the main or only source of animal protein. The true burden of declining fish resources falls upon the poorest and vulnerable groups who will be losing access to an important source of cheap protein and valuable micronutrients [87]. In addition, declining fish catches affect household income, thereby, affecting households' economic access to safe and nutritious food of their preferences.

It is now widely agreed that aquaculture production will meet the foreseen future increase in demand for fish through increased food production which enhances fish availability and job creation [14]. Aquaculture is one of the fastest-growing food-producing sectors but its contribution to SSA fish production is insignificant [31]. Adeleke et al. [31] state that some African governments have developed and adopted aquaculture-centered policies and strategic frameworks as an initiative to enhance aquaculture development. For example, in South Africa, the National Aquaculture Strategic Framework (NASF) was developed in 2012 to guide the sustainable development of the aquaculture sector through promoting good governance, provision of required support services, and investing in research and development to aid development of the sector. Troell et al. [88] argues that aquaculture development provides benefits to the environment by providing ecosystem services, such as bioremediation, waste removal, and habitat structure, alleviating pressure on wild stocks and replenishing depleted stocks.

The SSA aquaculture subsector faces several challenges that affect its total production, and these challenges include shortage of inputs (fish feed and fingerlings), high cost of feed, poor market infrastructure and access, inadequate human and financial resources, and weak governance [5,89]. For instance, SSA has few commercial fish feed producers to meet the local demand which has led to unregulated production of fish feed, for example, in Uganda, the inability of the fish feed industry to meet local fish feed demand and the absence of government regulation on fish feed quality assurance limits the development of aquaculture as fish farmers are not guaranteed of the quality of feed being used [90]. Obwanga et al. [91] argue that the growth of the aquaculture sector is positively correlated with the increased use of quality feed that meets the nutritional requirements of the cultivated species. Hence, low-quality fish feed deters aquaculture fish production. These challenges experienced by the aquaculture sector limit the potential of the sector to develop and increase fish supply into the region's market to meet the demand of the growing population. Thus, the ability of the aquaculture sector to significantly contribute to food and nutrition security is limited by the unavailability of inputs, human resources, and weak governance.

## 6. Implication for Food and Nutrition Policy

Despite the significant contribution of fish and fisheries to livelihoods and food and nutrition security, the fish and the fisheries sector has been long deserted from key food policy dialogues and associated funding [14,15]. For instance, the UN mainly focuses on agricultural systems such as resilient agricultural practices, land and soil quality, plant and livestock gene banks, agricultural subsidies, and access to land, to drive policy reforms and funding to meet the aims of its SDG 2 (end all forms of hunger and malnutrition) [92];the fisheries sector has not been specifically mentioned, nor do they offer specific guidance relevant to fish production systems [15]. Yet, empirical evidence from this study demonstrates that fish and the fisheries sector have the potential to significantly reduce hunger and malnutrition particularly in SSA.

Fish should be widely acknowledged as food rather than a natural resource and be included in food and nutrition policies. Including fish in food and nutrition policies acknowledges fish and fisheries as a food production system that might steer policy reforms and attract investments. Policymakers and development agencies should incorporate fish in policies aimed at improving overall food and nutrition security in SSA because fish is a rich source of animal protein and essential micronutrients. For instance, the National Agriculture Investment Program for Food and Security Nutrition 2018–2022 of Senegal clearly identified the fisheries sector as one of the key sectors needing investment to improve the food and nutritional status of the nation's population [93].

Furthermore, governments and development agencies should promote the consumption of fish to enhance the nutritional status of people, particularly the poor and marginalized groups in the region. Promoting fish and encouraging its consumption creates awareness of fish and its benefits to human health. Fish, particularly small indigenous fish species which are abundant in several African great lakes, can help the region reduce the high levels of undernutrition due to their high nutritional value. Studies by Longley et al. [94] and Marinda et al. [45] which focused on the contribution of fish to nutrition security in Zambia revealed that fish, mainly small indigenous species, improved the nutritional status, particularly in children, due to their high nutrient content of valuable micronutrients such as zinc, calcium, iodine, and iron, etc. Therefore, creating policies that promote and create awareness of the benefits of consuming fish can significantly contribute to increased fish consumption, consequently alleviating both under and malnutrition. In the midst of declining fish catches from wild capture fisheries, the aquaculture subsector in SSA is rapidly developing, contributing significantly to food and nutrition security and livelihoods of several thousands of people. Given the significance of the subsector, governments and development agencies should promulgate policies aimed at further developing the subsector. The policies should focus on ensuring access to factors of production and provision of fisheries and aquaculture extension services.

Lastly, policies should promote the conservation and management of fish resources to secure livelihoods and food and nutrition security of fishery-dependent societies. Fish resources are particularly relevant in SSA where the prevalence of undernutrition is high and the food and security situation worsening. Dwindling fisheries resources due to overexploitation, illegal unregulated fishing, and climate change [12,24,80] threaten food and nutrition security and livelihoods of the fishery-dependent groups. For example, a study by Muringai et al. [95] found that declining fish catches of fishers in Lake Kariba in Zimbabwe affected the fishing household's incomes, negatively affecting the ability of fishery-dependent households to buy various foods in desired quantities and preferences. Therefore, policies should promote the effective management and conservation of fishery resources without undermining the livelihoods of the resource users.

## 7. Conclusions and Recommendations

This review article sought to demonstrate the role of fish and the whole fisheries sector towards securing food and nutrition security in SSA. Sub-Saharan Africa was of particular interest in this review because of the reported high levels of undernourishment among its

inhabitants, the increasing demand for food to meet the region's rapidly growing population, and its endowment of vast fishery resources. Built on the available evidence, this review concludes that fisheries have great potential to contribute to the food and nutrition security status of SSA through the provision of a rich and cheap source of animal protein and valuable micronutrients, particularly for the most vulnerable and poor population. The sector supports livelihoods and generates income for millions of people in the SSA region. As a main or only source of protein for more than 30% of the people in SSA, fish should be recognized as food rather than only a natural resource and by doing so, policies can be shifted towards more integrated perspectives, moving beyond the simplistic productivism narratives to better consider how fish for food can be produced, distributed, and consumed in the region. Notwithstanding the growing recognition of fish as food, governments and research and development actors need to explicitly encourage the consumption of fish in human diets. The growing population and increasing demand of fish in an era of stagnant wild capture fish production provides an opportunity for the development of the aquaculture subsector to increase fish production and meet the growing demand of fish in the region.

Nevertheless, the fisheries sector in SSA is experiencing several challenges which include but are not limited to over exploitation of resources, declining wild capture fisheries stocks, climate change, shortage of inputs, poor market infrastructure, and access, inadequate human and financial resources, and weak governance. These challenges affect the ability of the sector to meet the increasing demand of fish, consequently, negatively affecting the food and nutrition status of the fishery-dependent households or communities. Governments, the private sector, research and development actors, and resource users should effectively manage the available stocks, invest in fisheries infrastructure, and strengthen governance of fisheries; only then can fish resources be protected and livelihoods and food and nutrition security of fishery-dependent groups secured. This research recommends future research on the role of policymakers, international organizations, cooperates, academics, and society in promoting the development of the fisheries sector as one of the key food production systems, to increase fish production and ensure sustainable fishing practices for future generations to enjoy the same benefits.

**Author Contributions:** Study conceptualization: R.T.M.; literature review R.T.M.; writing—original draft preparation: R.T.M.; writing—review and editing: R.T.M., P.M., R.T.L., R.M. and D.N.; and supervision: P.M. and R.T.L. All authors have read and agreed to the published version of the manuscript.

**Funding:** This research was funded by South Africa's National Research Foundation (NRF), grant number 86893.

**Institutional Review Board Statement:** Not applicable.

**Informed Consent Statement:** Not applicable.

**Data Availability Statement:** Not applicable.

**Acknowledgments:** The authors extend their sincere gratitude to Taetso Nkwagatse, Liboster Mwadzingeni, and Tatenda Ngwere for their support and assistance in literature search throughout the study.

**Conflicts of Interest:** The authors declare no conflict of interest.

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
