# Peer review of "Unlocking the Potential of Fish to Improve Food and Nutrition Security in Sub-Saharan Africa"

_sustainability, doi:10.3390/su14010318_

Round 1

Reviewer 1 Report

The authors discuss the role the fish sector could play in food and nutrition security in Sub-Saharan Africa. They acknowledge that the fish sector has been in some way isolated from food security and nutrition issues and the need to recognize that fish have the potential to significantly reduce hunger and malnutrition in SSA. Thus, they conclude that it would be desirable to include fish in food and nutrition policies.

The article is relevant and timely. Overall, the paper is straightforward, well-written and well structured. However, the article´s content tends to be at points way too general.

Revisions are needed in several aspects. I have some comments which I list below:

  • SSA comprise a lot of countries. Some are coastal countries, some are landlocked countries and some have great lakes. There must be a lot of heterogeneity between them in both consumption and production. Although the authors vaguely acknowledge this at some points (for example, in lines 291-292), this recognition is rather shallow. They limit themselves to provide a number of examples. I believe that the problems/challenges/opportunities must be very different between countries and that this fact should be emphasized more strongly.
  • I think the discussion on the trade-offs between the overexploitation of resources and the potential of fish to improve food and nutrition security should be further developed.
  • The same applies to the debate on the real potential of aquaculture: what role does aquaculture really play? And what role can it play in the longer term?
  • I would suggest being more specific on the last two sections —"Implication for Food and Nutrition policy” and “Conclusions and recommendations”. It is now a bit too general.
  • If possible, I would suggest illustrating somewhat more the part on “Sub-Saharan Africa’s food and nutrition status”, especially in relation to protein, fish and essential micronutrients consumption.

Reviewer 2 Report

I would suggest the authors to introduce a subchapter in which they present the danger to human health due to the consumption of contaminated fish, which have high concentrations of heavy metals, pesticides or hydrocarbons. The authors can give examples of articles that present such cases of contaminated fish in the studied area. Contaminated fish can have various effects on human health, but to prevent this, measures can be taken. These prevention measures, I think that already exist in the studied area and the authors can expose them.

Reviewer 3 Report

This is a very well-written and thought-provoking review article. The scope is manageable, the topic is very well-focused and relevant even for a broader audience. The title is clear, concise, and informative. The abstract should give an explanation about the fact that the authors wrote a summary article, to make it immediately clear to the reader - but it is only a suggestion. The map (Figure 1.) might not be needed, it is common knowledge what is Sub-Saharan Africa. Instead, try to add more statistical data, e.g. in Chapter 4.3. You might want to summarise the data mentioned there in a visually pleasing way - the readers would like it.

Round 2

Reviewer 1 Report

The paper has been improved significantly. However, I still have some comments which I include below:

  • The heterogeneity across SSA has been more clearly acknowledged now (for example, in lines 164-168). But it seems by the way it is written that all countries in SSA are endowed with rich fish resources. I was wondering if this is the case. I would suggest clarifying this point.
  • The section "Implication for Food and Nutrition policy” has been considerably improved. I appreciate especially the inclusion of the last paragraph. However, I still miss a bit more emphasis on the policy implications related to aquaculture and recommendations to promote the aquaculture sector.
  • I would suggest including a definition of nutrition security alongside the definition of food security. I think it could be useful in this particular case.
  • Lines 66-67: The text reads that “The aquaculture sector is rapidly growing because of the increasing fish demand”. Is it only because of the increasing fish demand?
  • Line 111: The number 433.3 does not match the exact number in Table 1.
  • Line 114: Number “239” does not match the “234.7” in Table 1.
  • Line 116: both did not report “approximately 118 million undernourished people in 2019".
  • Table 2: I would suggest adding the world´s average so as to be able to compare.
  • Table 3: I would suggest adding the world´s average so as to be able to compare more easily. I would also suggest explaining in more detail the data shown in the table.
  • Is there any risk to human health particular to SSA?
  • Line 322-324: This sentence is not clear from the text. I would suggest rewriting in a clearer way.
  • Even though the paper is overall well-written, some careful rewriting is needed before it is publishable.
